# Anti-Inflammatory Effects of 1,25(OH)2D/Calcitriol in T Cell Immunity: Does Sex Make a Difference?

**DOI:** 10.3390/ijms23169164

**Published:** 2022-08-15

**Authors:** Daniela Peruzzu, Maria Luisa Dupuis, Marina Pierdominici, Katia Fecchi, Maria Cristina Gagliardi, Elena Ortona, Maria Teresa Pagano

**Affiliations:** Center for Gender Specific Medicine, Istituto Superiore di Sanità, 00165 Rome, Italy

**Keywords:** vitamin D, sex hormones, gender, gender difference

## Abstract

Hypovitaminosis D is involved in various inflammatory, infectious and autoimmune diseases such as rheumatoid arthritis and multiple sclerosis. Moreover, the active form of vitamin D, calcitriol, has been shown to modulate the immune response, playing an anti-inflammatory effect. However little is known about the mechanisms underlying this anti-inflammatory effect and the potential sex differences of calcitriol immune regulation. Hence, the aim of this study was to investigate whether calcitriol could act differently in modulating T cell immunity of age-matched male and female healthy donors. We analyzed the effects of calcitriol in T lymphocytes from healthy women and men on the expression levels of the vitamin D receptor (VDR) and pro- and anti-inflammatory cytokine production. We showed that a treatment with calcitriol induced a significant increase in the VDR expression levels of activated T lymphocytes from male and female healthy subjects. Moreover, we found that calcitriol significantly reduced the expression level of pro-inflammatory cytokines IL-17, INF-γ and TNF-α in the T lymphocytes of both sexes. Notably, we observed that calcitriol induced a significant increase in the expression level of anti-inflammatory cytokine IL-10 only in the T lymphocytes from female healthy donors. In conclusion, our study provides new insights regarding the sex-specific anti-inflammatory role of calcitriol in T cell immunity.

## 1. Introduction

Vitamin D is a steroid hormone originating from cholesterol. The majority of vitamin D is produced by dermal synthesis after exposure to ultraviolet light B (UVB); however, a few foods naturally contain vitamin D (such as oily fish, cod liver oil, egg yolks and liver or organ meats) [1]. The two major types of vitamin D are vitamin D2 (ergocalciferol, from plants) and vitamin D3 (cholecalciferol, from animal sources) [2]. Both vitamin D3 and D2 are biologically inactive. They need a further enzymatic conversion to provide their active forms. First, vitamin D undergoes 25-hydroxylation in the liver to create 25-hydroxyvitamin D (25(OH)D), or calcidiol, the major circulating form of vitamin D, by 25-hydroxylase (CYP2R1) [2]. It is converted in the kidneys through 1-α-hydroxylation by 1α-hydroxylase (CYP27B1) to 1,25-dihydroxy vitamin D (1,25(OH)2D, or calcitriol), the most active form [3]. The role of calcitriol extends from mediating calcium homeostasis to functioning as a pluripotent hormone with various systemic effects [4,5]. The role played by calcitriol in immune modulation is well-known. Calcitriol, through vitamin D receptor (VDR) binding, exerts different actions on the innate and adaptive immune systems, among which are the suppression of inflammation and the promotion of tolerogenic responses [6,7].

Hypovitaminosis D adversely affects the bone mass, causing rickets in children and adolescents and osteoporosis and osteomalacia in adults. Hypovitaminosis D has also been linked to the onset/maintenance of cardiovascular diseases [8], chronic obstructive pulmonary disease [9], allergic asthma [10], type 2 diabetes [11] and autoimmune diseases [12,13,14]. Although contrasting data have been reported regarding sex differences in 25(OH)D plasma levels [15,16], there is evidence to suggest that gender-specific determinants could modulate 25(OH)D plasma concentrations [17]. In particular, time spent outdoors has been suggested to impact 25(OH)D levels mainly in women whereas physical activity and smoking affect 25(OH)D levels mainly in men [17].

In addition, sex differences in the immunomodulatory and anti-inflammatory effects of calcitriol have been suggested [5]. In particular, calcitriol has been observed in multiple sclerosis to induce a stronger inhibition of the production of pro-inflammatory cytokines and a higher induction of anti-inflammatory cytokine expression in lymphocytes from female patients in comparison with those from male patients [18]. Further supporting the sex-specific immune modulation of calcitriol, Spanier and co-workers [19] observed that calcitriol modulated regulatory T cell differentiation in an estrogen-dependent manner. Moreover, estrogen has been suggested to increase the expression level of the VDR in CD4+ T cells [20] and to decrease the expression of CYP24A1, the cytochrome P450 component of the 25-hydroxyvitamin D(3)-24-hydroxylase enzyme that inactivates calcitriol. In turn, calcitriol exerts a tissue-specific effect on peripheral estrogen metabolism [21]. However, whether calcitriol acts differently in modulating the T cell immunity of male and female healthy donors has not been deeply understood to date.

Hence, the aim of this study was to investigate the effective sex-specific role of calcitriol in modulating the immune response of healthy subjects, analyzing in T lymphocytes from males and females the effects of vitamin D3 on the expression levels of the VDR and cytokine production.

## 2. Results

### 2.1. Calcitriol Upregulates VDR Expression in Activated Peripheral Blood T Lymphocytes from Female and Male Healthy Donors

The VDR is a transcription factor and a member of the steroid hormone nuclear receptor family. After binding with 1,25(OH)2D, the VDR heterodimerizes with the retinoid X receptor (RXR), forming a complex with a high affinity for the vitamin D response element (VDRE) that modulates more than 900 genes involved in many physiological processes [22,23,24]. The VDR has been reported to be expressed in many tissues and in different cell types, including immune cells such as dendritic cells (DCs), macrophages/monocytes, neutrophils and T and B lymphocytes. Of note, resting monocytes and dendritic cells express an intracellular VDR whereas resting T and B lymphocytes express little to no VDRs. Accordingly, in our experiments we observed no or a low expression of the VDR in the resting T cells, as determined by a Western blot analysis (data not shown). After T lymphocyte activation with anti-CD3 mAb, we observed a significant increase in the VDR expression level induced by a treatment with calcitriol (treated versus untreated T cells, *p* =  0.0212 and *p* =  0.0117 in females and males, respectively, Figure 1). No significant difference was observed between the activated T cells from males and females.

### 2.2. Calcitriol Effects on Cytokine Production in Peripheral Blood T Lymphocytes from Female and Male Healthy Donors

As the data in the literature indicate that calcitriol is able to downregulate pro-inflammatory cytokine production in different cell types [25,26,27], we decided to analyze whether calcitriol could differently modulate pro-and anti-inflammatory cytokine production in the T lymphocytes from female and male healthy donors to highlight a possible sex-specific calcitriol role. A panel of pro-inflammatory (IFN-γ, TNF-α and IL-17) and anti-inflammatory (IL-4 and IL-10) cytokines was assessed. Calcitriol significantly reduced the intracellular expression level of pro-inflammatory cytokines in the T lymphocytes from both females and males (Figure 2A–C for IFN-γ, TNF-α and IL-17, respectively).

Notably, calcitriol was able to induce a significant increase in the intracellular expression level of the anti-inflammatory cytokine IL-10 only in the T cells from females (treated versus untreated cells, *p* = 0.0277, Figure 3A); no significant differences were observed in the IL-4 expression after the calcitriol treatment (data not shown).

## 3. Discussion

In this study, we analyzed the effects of calcitriol in modulating the expression levels of the VDR and the production of pro- and anti-inflammatory cytokines in the T lymphocytes from male and female healthy donors. We observed that the activated T lymphocytes under a treatment with calcitriol showed a significant increase in the expression levels of the VDR in both female and male healthy donors. The sex-independent effect of calcitriol on the VDR expression that we observed in our study was in contrast with the data reported by Cheema and co-workers [20], who reported a role of estrogen in increasing the VDR expression in T cells. The mean age (50 years) of the healthy women enrolled in our study could explain why we did not observe an estrogen-mediated effect.

After a treatment with calcitriol, we also found a significant reduction in the expression level of pro-inflammatory cytokines (i.e., IL-17, INF-γ and TNF-α) in the T cells from both the female and male healthy donors. In contrast with our data, Correale and co-workers, analyzing sex differences in the immunomodulatory and anti-inflammatory effects of calcitriol in T lymphocytes from patients with multiple sclerosis, observed that calcitriol reduced the expression of pro-inflammatory cytokines only in the T cells of female patients. The different activation status of the T lymphocytes of the healthy subjects in our study from the patients with multiple sclerosis in the article of Correale et al. might explain these different results. Notably, and accordingly with that observed by Correale and co-workers [18], we observed that calcitriol induced a significant increase in the expression level of the anti-inflammatory cytokine IL-10 only in the healthy female subjects.

Overall, our data suggested that calcitriol, by lowering pro-inflammatory cytokine production in the T cells from the female and male healthy subjects, was able to reduce the inflammatory status independent of the sex. However, calcitriol, which induced the expression of IL-10 only in the T cells from the females, seemed to prevent the onset of inflammation in a sex-specific manner. Further and deeper analyses will clarify whether the increase in IL-10 production could be responsible for the sex differences in the anti-inflammatory effects of calcitriol. Moreover, which T cell subset is responsible for the increased IL-10 production should also be investigated to better clarify the mechanisms underlying the sex differences in the vitamin D effects.

Several studies have suggested the existence of a cross-talk between estrogen and vitamin D [28,29]. In particular, women accepting exogenous estrogen showed a significant increase in plasma 25(OH)D levels [30] and low 25(OH)D levels were associated with low E2 levels in women [31].

Our data supported the notion that calcitriol is also able to play a direct role in determining a sex-specific modulation of inflammation, independent of an estrogen effect on its concentration. However, to validate this observation, further studies with a larger number of pre- and post-menopausal women and age-matched males are needed as well as investigations into plasma estradiol concentrations.

A limitation of our study was the failure to measure several gender-related factors such as the use of vitamin D supplementation and the use of estrogen therapies in healthy participants.

In conclusion, our study provides new insights regarding a sex-specific anti-inflammatory role of calcitriol in T cell immunity. Further studies are needed to clarify the sex-specific mechanisms underlying the anti-inflammatory role of calcitriol to direct its preventive or therapeutic use in the most appropriate way, also in consideration of sex and age.

## 4. Materials and Methods

### 4.1. Study Population

Thirty healthy blood donors were enrolled by the Immunohaematology and Transfusion Unit of Umberto I Polyclinic, Rome, Italy (15 females, mean age 50, range 20–70 years and 15 males, mean age 48, range 24–65 years). Informed consent was obtained from all donors before collecting the blood samples.

### 4.2. Isolation of Peripheral Blood Lymphocytes and Cell Culture Conditions

Peripheral blood mononuclear cells (PBMCs) were isolated by Ficoll–Hypaque density-gradient centrifugation (Lympholyte-H; Cedarlane, Paletta Court, ON, Canada). The separation of untouched T cells from the PBMCs was performed by an immunomagnetic-based depletion of non-T cells using a Pan T Cell Isolation Kit II (Miltenyi Biotec, Bergisch Gladbach, Germany). The cells were cultured in an RPMI-1640 medium (Gibco BRL, Grand Island, NY, USA) supplemented with 10% FBS, 2 mM glutamine (Sigma, St. Louis, MO, USA) and 50 µg/mL gentamycin (Sigma) at 37 °C in a humidified 5% CO_2_ atmosphere. Calcitriol (Sigma) was dissolved in ethanol and diluted in the treatment medium at the required concentrations. For the lymphocyte activation, the cells were cultured in the presence of a plate-bound anti-CD3 monoclonal antibody (mAb, clone UCHT1, R & D Systems, Minneapolis, MN, USA) at 4 µg/mL for 72 h and treated with calcitriol for the last 48 h of culture at a dose of 100 nM. For the cytokine analysis, the untreated or treated cells were stimulated as follows: (i) for the IFN-γ, TNF-α and IL-4 analyses: 25 ng/mL phorbol myristate acetate (PMA, Sigma) and 1 µg/mL ionomycin (Sigma) for the last 16 h of culture; (ii) for the IL-17 analysis: 50 ng/mL PMA (Sigma) and 1 µg/mL ionomycin (Sigma) for the last 4 h of culture; and (iii) for IL-10: 2.5 µg/mL phytohemagglutinin (PHA, Sigma) for the last 16 h of culture. To inhibit the cytokine secretion, 10 µg/mL brefeldin A (Sigma) was added to each condition at the beginning of the stimulation.

### 4.3. SDS-PAGE and Western Blot

The T cells were lysed in a RIPA buffer (100 mM tris(hydroxymethyl)aminomethane (Tris)-HCl pH 8, 150 mM NaCl, 1% Triton X-100, 1 mM MgCl_2_) in the presence of a complete protease-inhibitor mixture (Roche Diagnostics GmbH, Mannheim, Germany). The protein content was measured by a Bradford assay (Bio-Rad Laboratories, Richmond, CA, USA). After the SDS-PAGE, the proteins were transferred onto a nitrocellulose membrane (GE Healthcare, Pittsburgh, PA, USA) by means of a Trans-Blot transfer cell (Bio-Rad Laboratories). The membranes were then blocked in 5% non-fat milk and incubated with the proper antibodies in Tris-buffered saline (TBS) containing 0.1% Tween 20 and 5% non-fat milk. A mouse anti-human VDR (clone D6, Santa Cruz Biotechnology, Santa Cruz, CA, USA) was used as the primary antibody. Peroxidase-conjugated goat anti-mouse IgG (Bio-Rad Laboratories) was used as a secondary antibody and the reaction was developed using an ECL Prime Western Blotting Detection Reagent (GE Healthcare). To ensure the presence of equal amounts of protein, the membranes were re-proved with rabbit anti-human glyceraldehyde 3-phosphate dehydrogenase Ab (GAPDH, Sigma). The quantification of the protein expression was performed by a densitometry analysis of the autoradiograms (GS-700 Imaging Densitometer, Bio-Rad Laboratories).

### 4.4. Flow Cytometry

Cell surface phenotyping was performed by flow cytometry, as previously described [32]. Allophycocyanin (APC)-conjugated anti-CD3 mAbs (BD Biosciences, San Jose, CA, USA) and an equal amount of the mouse IgG isotype control were used in parallel. An analysis of the cytokine production at a single-cell level was performed. Briefly, the treated cells were either fixed with 4% paraformaldehyde and permeabilized with a FACS permeabilizing solution (BD Biosciences) for IFN-γ, TNF-α, IL-4 and IL-10 detection or were fixed and permeabilized with an intracellular fixation and permeabilization buffer (eBioscience, San Diego, CA, USA) for IL-17 detection. The following cytokine-specific mAbs were used: fluorescein isothiocyanate (FITC)-labeled anti-IFN-γ, PE-labeled anti-IL-4, PE-labeled anti-IL-10 (all from BD Biosciences), PE-labeled anti-TNF-α and FITC-labeled anti-IL-17A (eBioscience). Appropriate isotypic negative controls were run in parallel. To determine the frequency of the T cells, the total lymphocytes were first gated by a forward and side scatter and then additionally gated for the CD3 molecule expression. The acquisition was performed on a FACSCalibur flow cytometer (BD Biosciences) and at least 50,000 events per sample were run. The data were analyzed using Cell Quest Pro software (BD Biosciences).

### 4.5. Statistical Analysis

The statistical analysis was performed with a Mann–Whitney U test using the statistical package Prism 9 (GraphPad Software, San Diego, CA, USA). Any *p*-values lower than 0.05 were considered to be significant.

## Figures and Tables

**Figure 1 ijms-23-09164-f001:**
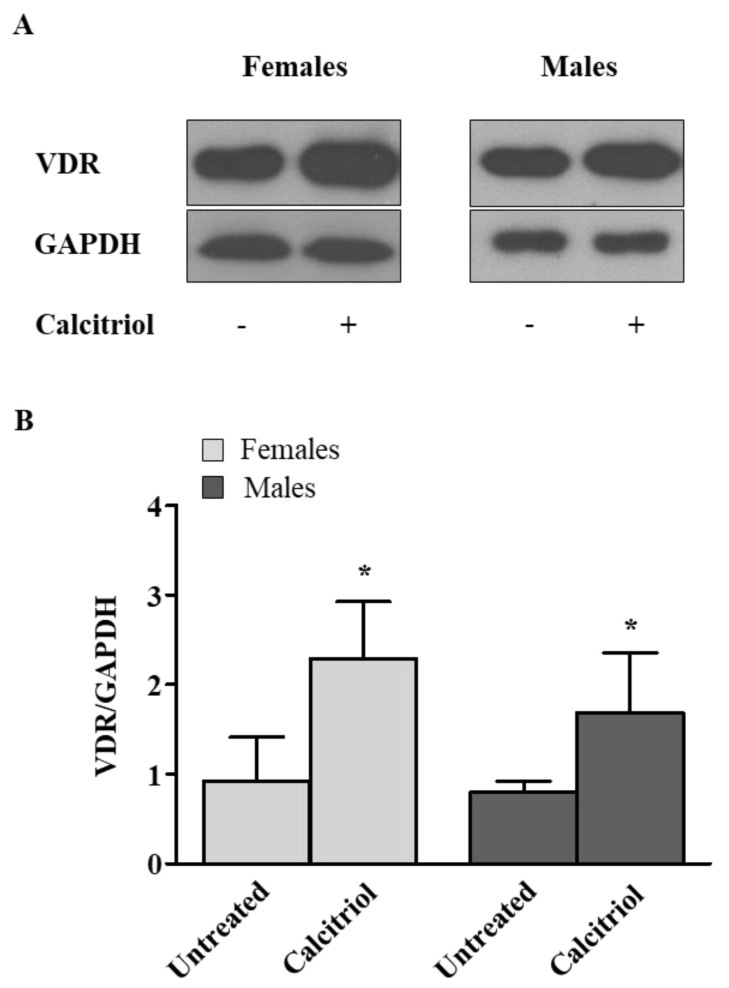
Calcitriol upregulates VDR expression in activated peripheral blood T lymphocytes from female and male healthy donors. VDR levels were evaluated by Western blot analysis of T cell lysates after 48 h of calcitriol treatment. (**A**) Blots shown are representative of experiments performed in T cells from 10 randomly selected female and male healthy subjects. (**B**) Densitometry analysis of VDR levels relative to glyceraldehyde 3−phosphate dehydrogenase (GAPDH) is shown. Results are shown as mean ± SD. * *p* < 0.05 versus untreated cells.

**Figure 2 ijms-23-09164-f002:**
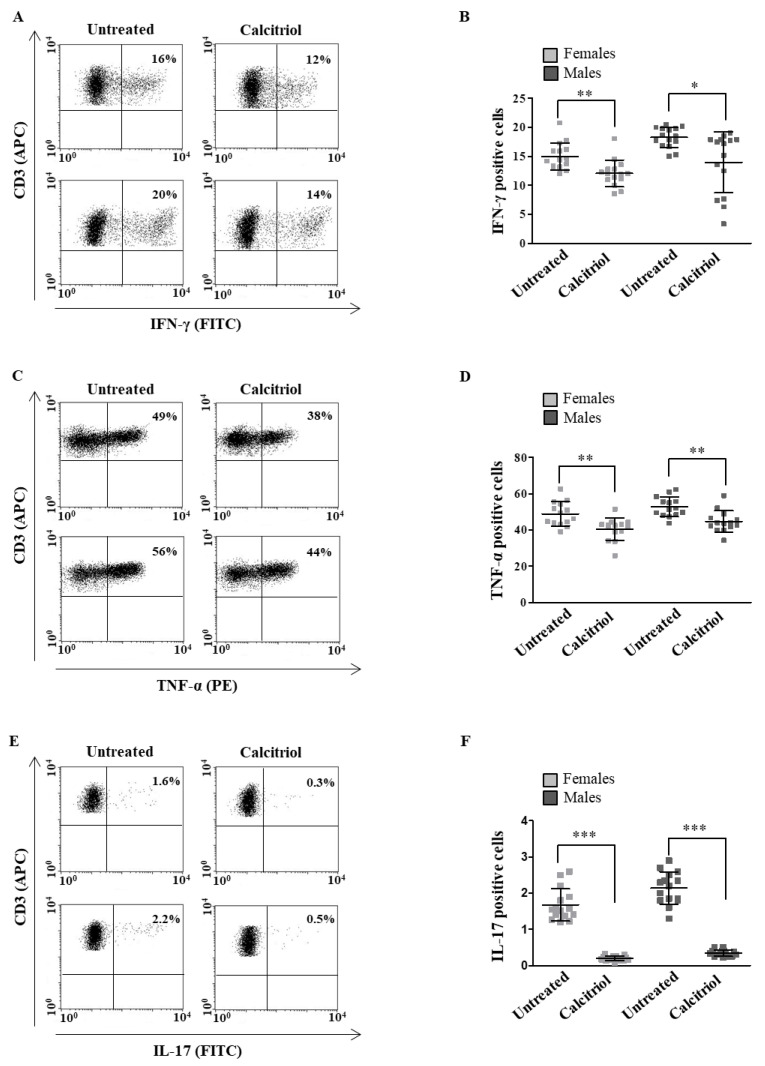
Calcitriol effects on pro-inflammatory cytokine production in peripheral blood T lymphocytes from female and male healthy donors. Cytokine expression was analyzed in 15 female and 15 male healthy subjects by flow cytometry after 48 h of culture with calcitriol and stimulation with phorbol myristate acetate (PMA) and ionomycin in the presence of brefeldin A for the last 16 h of culture, as detailed in Section 4. Results from representative female (upper panels) and male (lower panels) healthy donors are shown (**A**,**C**,**E**). Data are also reported as mean  ±  SD (**B**,**D**,**F**). * *p* < 0.05, ** *p* < 0.01, *** *p* < 0.001 versus untreated cells.

**Figure 3 ijms-23-09164-f003:**
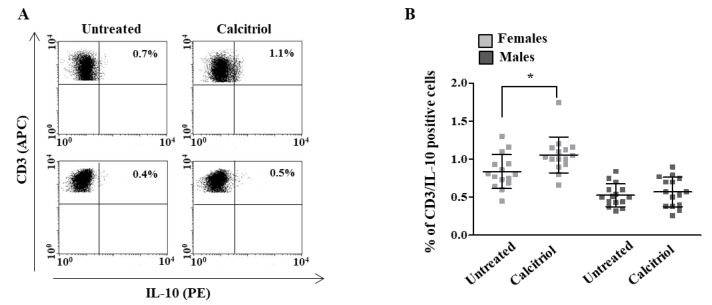
Calcitriol effects on anti-inflammatory cytokine production in peripheral blood T lymphocytes from female and male healthy donors. IL-10 expression was analyzed in 15 female and 15 male healthy subjects by flow cytometry after 48 h of culture with calcitriol and stimulation with phytohemagglutinin (PHA) in the presence of brefeldin A for the last 16 h of culture, as detailed in Section 4. (**A**) Results from representative female (upper panels) and male (lower panels) healthy donors are shown. (**B**) Data are also reported as mean ± SD. * *p* < 0.05 versus untreated cells.

## Data Availability

The data supporting this study’s findings are available upon request.

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
