# Peer review of "Anti-Inflammatory Effects of 1,25(OH)2D/Calcitriol in T Cell Immunity: Does Sex Make a Difference?"

_ijms, 2022, doi:10.3390/ijms23169164_

Round 1

Reviewer 1 Report

The authors are concerned with the metabolic intermediate of Vitamin D2 and D3 1,25(OH)2D also known as calcitriol. The main role of calcitriol is in mediating calcium homeostasis. In hypovitaminosis this intermediate adversely affects bone mass. The activities that can lead to this disease are said to affect both genders to different degrees. Specifically, the authors focus on the sex differences in the role of calcitriol on immunomodulatory and anti-inflammatory effects. In women, researchers have observed immune modulation in the presence of calcitriol scaled with estrogen concentration. This could be attributed to estrogens' role in increasing the expression of VDR in CD4+ T cells and decreasing the expression of CYP24A1.  VDR is a calcitriol binding transcription factor that forms a heterodimer with retinoid X receptor following binding to calcitriol. The authors evaluated the role of calcitriol on VDR expression following activation of T cells with anti-CD3 mAb. In this experiment, calcitriol treatment resulted in increased VRD expression following activation when compared to the control. Next, the authors evaluated pro and anti-inflammatory cytokine production on peripheral blood T lymphocytes following activation in the presence of calcitriol. In this study, calcitriol reduced the expression of pro-inflammatory cytokines equally in both genders. The only observable cytokine expression dependent on sex was IL-10 which was significantly increased in females.

One strong point behind the author's results is that they are supported by in vitro results that were conducted in other laboratories. This connection is strengthened further by the well-established connection between calcitriol and estrogen. Furthermore, all their conclusions from the data had some degree of observable significance. Another strong point is their studies covered a large range of ages and the number of males and females was kept consistent.

One of the major weaker points is the lack of demographic information. I feel that the evidence indicates that estrogen concentration is attributed to the change in calcitriol effects. As the authors point out in the conclusion the expression of estrogen can change in the female population depending on age. I think the authors should group the female population into different age groups to see if there is any difference in calcitriol treatment. Furthermore, I would suggest that the authors could benefit from increasing the number of participants.

Overall, this publication highlights an interesting effect of calcitriol on the immune response between both genders and adds to previous studies conducted in other labs that connect calcitriol activity with human patients.

Author Response

We thank the Reviewer for his/her comments. We agree that a major criticism of this work is the lack of demographic information. We have enrolled 5 females  under 45 (pre-menopausal) and 6 over 55 (post-manopausal) whereas 4 females are 45-55 years old i.e., in peri-menopausal period.  We will plan a further longitudinal study enrolling  larger number of pre-and post-menopausal females  and age -matched males also evaluating plasma estradiol concentration.

We specified this in Discussion and revised English language

Reviewer 2 Report

How is the age distribution for each group even though the average age is similar? As we know, a lot of factors infect the immune cells behavior, such as diet, stress, age, supplement Intake. How do you remove the biases?

What about the cell viability for trated or untreated cells?

The results seems not promising. I noticed that the SD of female group from figure 3B is quite big. I am wondering is there any outliar? What is the distribution for each single individual in female group? Maybe a distribution level of figure will make more sense. And have you test other anti-inflammatory cytokine secretion?

More references are needed in the first paragraph. And more references in L158 have to cite. 

I have to say only 15 participants for each group is too small to make the conclusion. 

English language must improve.

L14, it should be the aim of this study. 

25-hydroxyvitamin D (25(OH)D) should be listed in the first place of 25(OH)D appearing. For 1,25(OH)2D as well. 

delete L87, figure 1. same as L121, L105

Author Response

R: How is the age distribution for each group even though the average age is similar? As we know, a lot of factors infect the immune cells behavior, such as diet, stress, age, supplement Intake. How do you remove the biases?

We have enrolled 5 females  under 45 (pre-menopausal) and 6 over 55 (post-menopausal) whereas 4 females are 45-55 years old i.e., in peri-menopausal period. males are age-matched. As we stated in Discussion  "A limitation of our study is the failure to measure some gender-related factors, such as the use of vitamin D supplementation and the use of estrogen therapies in healthy participants." Diet and stress are other factors that we were not able to measure and only a study with a larger number of samples could avoid their biased effects. 

R: What about the cell viability for trated or untreated cells?

We did not observe any difference in cell viability in treated and untreated cells.

R: The results seems not promising. I noticed that the SD of female group from figure 3B is quite big. I am wondering is there any outliar? What is the distribution for each single individual in female group? Maybe a distribution level of figure will make more sense. And have you test other anti-inflammatory cytokine secretion?

We changed the graphs as requested by reviewer.

We tested also IL-4 but no differences was observed (data not shown) as reported in the manuscript.

R: More references are needed in the first paragraph. And more references in L158 have to cite. 

DONE

R: I have to say only 15 participants for each group is too small to make the conclusion. 

Yes, we agree that this is a limitation of our study and we specify this in Discussion. However, we found  significant differences that could be suggestive.

R. English language must improve.

We improve English language

R: L14, it should be the aim of this study. 
yes we changed accordingly

R: 25-hydroxyvitamin D (25(OH)D) should be listed in the first place of 25(OH)D appearing. For 1,25(OH)2D as well. 

DONE

R: delete L87, figure 1. same as L121, L105

DONE

Reviewer 3 Report

In a general reappraisal of the effects of vitamin D supplementation, this manuscript adds a little piece of information on sex-difference in the anti-inflammatory effects of vitamin D.

The bottom line of this manuscript is that, upon treatment with vitamin D, PBL from female subjects express IL-10, while those from males fail to do this. This expression may explain sex differences in the anti-inflammatory effects of vitamin D. However, no attempt is made to assess if the increased production of IL-10 is responsible for those differences, neither is defined which T-cell subset is responsible for. Altogheter, this (small) difference in IL-10 production can be important, but this is not the subject of this report. 

Minor comments:

I would correct the second part of title “…sex makes a difference?” in “…does sex make a difference?”

In the Abstract, VDR is not defined.

Is it not specified if the normal subjects included in the study were assuming supplemental vitamin D, neither exclusion of these subjects is planned in the Inclusion criteria.

Author Response

We thank the Reviewer for his/her comments. We agree that a limitation of this brief report is ththe lack of a  deeper analysis to clarify whether the increase IL-10 production could be responsible for sex differences in the anti-inflammatory effects of calcitriol. Moreover, which T-cell subset could be responsible for IL-10 increased production should be also investigated for better clarify the mechanisms underlying sex differences in vitamin D effects. Now we specified these limitations of the study in the Discussion.

Minor comments:

R: I would correct the second part of title “…sex makes a difference?” in “…does sex make a difference?”

DONE

R: In the Abstract, VDR is not defined.

DONE

R: Is it not specified if the normal subjects included in the study were assuming supplemental vitamin D, neither exclusion of these subjects is planned in the Inclusion criteria.

In Discussion we stated that "A limitation of our study is the failure to measure some gender-related factors, such as the use of vitamin D supplementation and the use of estrogen therapies in healthy participants."

Round 2

Reviewer 2 Report

I think the author can add the age distribution for females and males into method part or discussion part to help the reader understand for reducing the biases. 

Reviewer 3 Report

The Authors have modified the paper according to suggestions. No further change is required by this Reviewer.